# MambaSIC: Mamba-based Stereo Image Compression with Bi-directional Multi-reference Entropy Model

## Abstract

Stereo image compression (SIC) has become increasingly vital with its applications surging in fields such as 3D reconstruction and autonomous navigation. Previous methods leverage cross-attention to model inter-view redundancy and employ autoregressive entropy models to predict probability distributions, achieving impressive rate-distortion performance. However, they suffer from slow coding speed due to the quadratic complexity of cross-attention mechanisms and the spatial autoregressive iterations of the entropy models. To address these limitations, we propose MambaSIC, which introduces two key innovations. First, we propose a Mamba-based stereo visual state space block (stereo VSSB) that leverages its linear complexity and long-range modeling capabilities to more rapidly and efficiently capture redundancy information between the two views. Second, to accelerate the compression process and enhance the accuracy of probability distribution estimation, we introduce a bi-directional multi-reference entropy model that utilizes a checkerboard partitioning strategy and the stereo VSSB to get rich inter-view priors. Experimental results demonstrate that our MambaSIC outperforms the state-of-the-art methods in both rate-distortion performance and coding efficiency. Moreover, it achieves the smallest inter-view PSNR discrepancy, resulting in more balanced reconstruction quality.

## 1 Introduction

Stereoscopic image processing leverages binocular vision to simulate the human ability of perceiving depth and creating a holographic viewing. This technique plays a crucial role in applications such as virtual reality (Fehn, 2004), autonomous navigation (Duba et al., 2024), and 3D reconstruction (Fujimura et al., 2018), therefore resulting in a surging demand for efficient transmission and storage of high-quality stereo images in recent years. This underscores the importance of stereo image compression (SIC), which aims to reduce storage overhead without compromising visual quality.

Stereo images, presenting content captured from two different viewpoints, exhibit strong inter-view correlations and provide critical spatial information. Traditional stereo image compression methods, such as MVC (Vetro et al., 2011) and MV-HEVC (Tech et al., 2015), use predictive coding, where one view serves as a reference to estimate the other view, and the estimation differences are encoded. However, these methods depend on handcrafted prediction modules, which struggle to effectively capture intricate inter-view correlations in complex scenes. Recent learning-based single-image compression methods (Ballé et al., 2017; 2018; Jiang & Wang, 2023) have made notable progress by introducing advanced nonlinear transforms and entropy models, motivating the application of deep learning to stereo image compression. Early efforts to apply convolutional neural networks (CNNs) and hyperprior models in stereo image compression primarily relied on dense distortion fields (Liu et al., 2019; Zhai et al., 2022) or rigid homography transformations (Deng et al., 2021; 2023) to model disparity. While efficient, their performance is constrained by limited receptive fields and simplistic entropy models. Moreover, unidirectional frameworks often cause imbalanced reconstruction quality between views. Recent advances (Liu et al., 2024c; Zhang et al., 2023) leverage cross-attention and bi-directional autoregressive entropy models to improve rate-distortion performance, but at the cost of significantly increased computational complexity. As illustrated in Fig. 1, achieving high compression performance with reduced coding time remains a critical challenge.

Recently, Mamba has demonstrated stronger global modeling capability than attention mechanisms in vision tasks (Liu et al., 2024b; Qin et al., 2024) while maintaining linear complexity, pointing to a promising direction for improvement. However, its inability to capture inter-view correlations and its limited local modeling capacity hinder its application to SIC. To address the above limitations, we propose stereo visual state space block (stereo VSSB), which enables both local and global context transfer across views. In stereo VSSB, we enhance the local and global features of the two views using CNN-based networks and the stereo visual state space layer (stereo VSSL), respectively. Within stereo VSSL, the stereo gating mechanism and cross-view matrix capture inter-view redundancy. This design avoids the quadratic complexity of cross-attention while fully exploiting Mamba's strengths in long-range dependency modeling and representation learning.

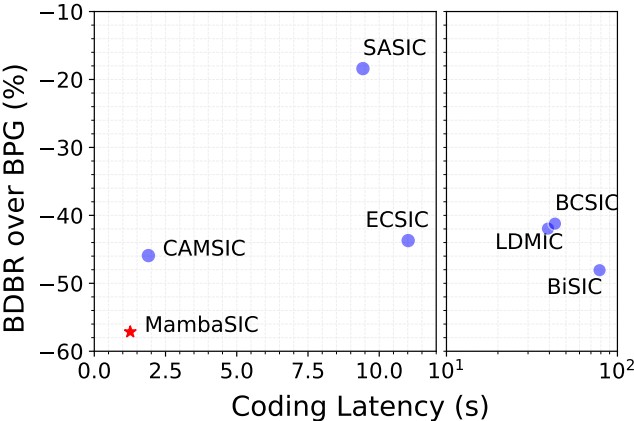

Figure 1: BDBR for PSNR (lower is better) vs coding latency on Instereo2K (Bao et al., 2020). MambaSIC achieves the best trade-off between compression performance and latency.

In addition to the type of the neural network, the design of entropy model is also an important technique in SIC. We develop a bi-directional multi-reference entropy model to accelerate entropy coding and enhance contextual conditioning. Our entropy model adopts a checkerboard pattern to partition latent representations, enabling it to achieve remarkable inference efficiency compared to spatial auto-regressive iterations (Lei et al., 2022; Liu et al., 2024c). Notably, instead of adopting convolution or attention modules in previous methods, we adopt stereo VSSB to fuse the information of left-view priors and right-view priors to generate abundant inter-view priors, which effectively exploits the correlation between stereo views and enhances the probability estimation for entropy coding. The entire procedure is designed to be fully symmetric and bidirectional, preventing significant quality discrepancies between the reconstructed left and right view images.

Building on the above improvements, we propose MambaSIC, a powerful and efficient stereo image compression framework that achieves an optimal balance between efficacy and efficiency. In summary, our contributions are as follows:

- We design a Mamba-based stereo context transfer module, stereo VSSB, as non-linear transform to better eliminate redundancy between the stereo views while maintaining linear complexity.

- We introduce a bi-directional multi-reference entropy model that leverages a spatial checkerboard pattern and the stereo VSSB to achieve efficient and compact entropy coding.

- On standard benchmark datasets, MambaSIC surpasses current state-of-the-art SIC baselines in both compression performance and speed, while also achieving more balanced reconstruction quality with the smallest inter-view PSNR discrepancy.

## 2  RELATED WORK

### 2.1  STEREO IMAGE COMPRESSION

Single image compression performs poorly on stereo images because it ignores inter-view correlations. This motivates stereo image compression research. Traditional methods, such as MVC (Sullivan et al., 2012) and MV-HEVC (Tech et al., 2015), use hand-crafted disparity compensation. Recent learning-based methods improve performance and fall into unidirectional and bi-directional coding. Unidirectional methods (Deng et al., 2021; 2023; Liu et al., 2019; Wödlinger et al., 2022; Zhai et al., 2022) predict a disparity-compensated view and encode residuals to reduce redundancy. Bi-directional methods (Lei et al., 2022; Liu et al., 2024c) use cross-attention to exploit mutual information and balance quality. Other works (Huang et al., 2023; Mital et al., 2023; Zhang et al., 2023; Xia et al., 2023) explore distributed multi-view coding with independent encoders and a joint decoder. However,

all above SIC methods primarily rely on convolutional networks or cross-attention mechanisms for alignment, which fails to capture long-distance spatial dependencies. Therefore, we explore to apply the Mamba architecture in stereo image matching.

## 2.2 VISUAL STATE SPACE MODEL

State Space Models (SSMs) are efficient alternatives to Transformers for sequence modeling, with linear complexity in capturing long-range dependencies. Recent works like S4 (Gu et al.), S5 (Smith et al.), and Mamba (Gu & Dao, 2023) improve SSM architectures and achieve strong results across domains. This drives interest in applying SSMs to vision, where spatial structures also show sequential dependencies. Vision Mamba (Zhu et al.) uses bi-directional scanning for inter-patch relations, and VMamba (Liu et al., 2024b) extends it with four-directional scanning. SSMs are also applied to segmentation (Xing et al., 2024; Zhang et al., 2024a), super-resolution (Guo et al., 2024; Xiao et al., 2024), and remote sensing (Chen et al., 2024; Liu et al., 2024a), offering lower cost with strong performance. Building on this progress, we introduce Mamba into the field of stereo image compression and propose a Mamba-based stereo matching method.

## 3 PROPOSED METHOD

### 3.1 PROBLEM FORMULATION

Fig. 2(a) shows the network architecture. Given stereo images $\boldsymbol{x}_l, \boldsymbol{x}_r \in \mathbb{R}^{3 \times H \times W}$, the encoder $g_a$ produces latent representations $\boldsymbol{y}_l, \boldsymbol{y}_r \in \mathbb{R}^{M \times \frac{H}{16} \times \frac{W}{16}}$. These are quantized to $\hat{\boldsymbol{y}}_l, \hat{\boldsymbol{y}}_r$. The joint decoder $g_s$ then reconstructs stereo images $\hat{\boldsymbol{x}}_l, \hat{\boldsymbol{x}}_r$. Since quantizer $Q$ is non-differentiable, we use mixed quantization (Minnen & Singh, 2020) in training. Latents are perturbed with uniform noise for bitrate estimation, while rounded latents use straight-through gradients for reconstruction. The compression process can be written as follows:

$$\begin{aligned}
\boldsymbol{y}_l, \boldsymbol{y}_r &= g_a(\boldsymbol{x}_l, \boldsymbol{x}_r; \phi), \\
\hat{\boldsymbol{y}}_l &= Q(\boldsymbol{y}_l), \ \hat{\boldsymbol{y}}_r = Q(\boldsymbol{y}_r), \\
\hat{\boldsymbol{x}}_l, \hat{\boldsymbol{x}}_r &= g_s(\hat{\boldsymbol{y}}_l, \hat{\boldsymbol{y}}_r; \theta).
\end{aligned} \tag{1}$$

where $\phi$ and $\theta$ are learnable parameters of the encoder $g_a$ and decoder $g_s$.

To reduce the statistical redundancy of the quantized representation $\hat{\boldsymbol{y}}_l, \hat{\boldsymbol{y}}_r$ by entropy coding, each element $\hat{y}_{l,i}, \hat{y}_{r,i}$ is modeled as a univariate Gaussian random variable with mean $\mu_{l,i}, \mu_{r,i}$ and standard deviation $\sigma_{l,i}, \sigma_{r,i}$, where $i$ denotes the position of each element in a vector-valued signal. We propose a bi-directional multi-reference entropy model to predict the probability distribution parameters $\boldsymbol{\mu}_l, \boldsymbol{\sigma}_l$ and $\boldsymbol{\mu}_r, \boldsymbol{\sigma}_r$, with more details provided in Section 3.3.

### 3.2 STEREO CONTEXT TRANSFER WITH VISUAL STATE SPACE

The core challenge in SIC lies in effective transfer of the shared information between the two views. To address this, we propose methods that focus on three key aspects: the utilization of local and global information, the information fusion within the gated network, and the state update process in the state space model. These are discussed in Sections 3.2.1, 3.2.2, and 3.2.3, respectively.

### 3.2.1 STEREO VISUAL STATE SPACE BLOCK

Mamba (Gu & Dao, 2023) has a larger receptive field than Transformers and captures information from distant regions. (Liu et al., 2023) shows that combining local and global information improves performance. Based on this, we design the Stereo Visual State Space Block (Stereo VSSB), which uses Mamba for global information transfer and convolution for local information transfer.

The structure of our Stereo VSSB module is illustrated in Fig. 2(b). The input stereo features $\boldsymbol{f}_l, \boldsymbol{f}_r \in \mathbb{R}^{N \times H_f \times W_f}$ first pass through a $1 \times 1$ convolutional layer without changing the channel dimension. Next, the convolved features are then split along the channel dimension into $\boldsymbol{f}_l^{\text{Local}}, \boldsymbol{f}_l^{\text{Global}}, \boldsymbol{f}_r^{\text{Local}}, \boldsymbol{f}_r^{\text{Global}} \in \mathbb{R}^{\frac{N}{2} \times H_f \times W_f}$, respectively. Through this operation, the local and global features of the left and right views are separated and transferred individually. Then we

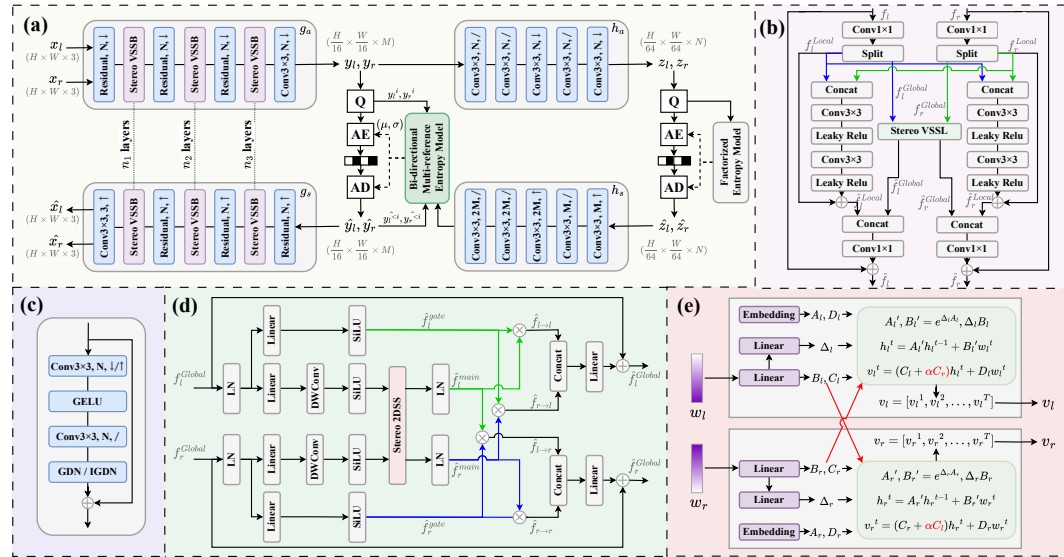

Figure 2: (a) Overall architecture of our proposed method MambaSIC. AE and AD are arithmetic encoder/decoder for entropy coding. Q denotes quantization. (b) Stereo Visual State Space Block, (c) Residual block, (d) Stereo Visual State Space Layer and (e) Stereo 2D Selective Scan. The blue and green lines represent features extracted from the left view and right view, respectively. The red line indicates that the matrix $C$ from another view is weighted and integrated into the current view.

concatenate the two local features $\boldsymbol{f}_l^{\text{Local}}, \boldsymbol{f}_r^{\text{Local}}$ sequentially and process them as follows:

$$
\begin{aligned}
\hat{\boldsymbol{f}}_l^{\text{Local}} &= \text{CLR}(\text{Cat}(\boldsymbol{f}_l^{\text{Local}}, \boldsymbol{f}_r^{\text{Local}})) + \boldsymbol{f}_l^{\text{Local}}, \\
\hat{\boldsymbol{f}}_r^{\text{Local}} &= \text{CLR}(\text{Cat}(\boldsymbol{f}_r^{\text{Local}}, \boldsymbol{f}_l^{\text{Local}})) + \boldsymbol{f}_r^{\text{Local}},
\end{aligned}
\tag{2}
$$

where CLR denotes a network composed of convolutional layers and leaky ReLU activations. For the global features $\boldsymbol{f}_l^{\text{Global}}, \boldsymbol{f}_r^{\text{Global}}$, they are input into a stereo visual state space layer (discussed in Section 3.2.2) and obtain the fusion features $\hat{\boldsymbol{f}}_l^{\text{Global}}, \hat{\boldsymbol{f}}_r^{\text{Global}}$. Finally, we concatenate the local and global features along the channel dimension and pass them through a $1 \times 1$ convolution to fuse local and non-local information. A skip connection is used between these combined/fused features and the input features $\boldsymbol{f}_l, \boldsymbol{f}_r$. This process is expressed as follows:

$$
\begin{aligned}
\hat{\boldsymbol{f}}_l &= \text{Conv}_{1\times1}(\text{Cat}(\hat{\boldsymbol{f}}_l^{\text{Local}}, \hat{\boldsymbol{f}}_l^{\text{Global}})) + \boldsymbol{f}_l, \\
\hat{\boldsymbol{f}}_r &= \text{Conv}_{1\times1}(\text{Cat}(\hat{\boldsymbol{f}}_r^{\text{Local}}, \hat{\boldsymbol{f}}_r^{\text{Global}})) + \boldsymbol{f}_r.
\end{aligned}
\tag{3}
$$

The Stereo VSSB are inserted after the first three downsampling blocks in the encoder $g_a$ and the first three upsampling blocks in the decoder $g_s$, as shown in Fig. 2(a). This placement ensures the complementary fusion of the left and right perspective information across multiple dimensions.

### 3.2.2 STEREO VISUAL STATE SPACE LAYER

Selective state space and gating are two key parts of Mamba (Gu & Dao, 2023). The first enables information interaction, and the second controls information flow. Based on them, we propose a stereo 2D selective scan and stereo gating connection to control information transfer across dimensions, as shown in Fig. 2(d).

Specifically, the global features $\boldsymbol{f}_l^{\text{Global}}, \boldsymbol{f}_r^{\text{Global}}$ are first passed through layer normalization and a linear layer, after which they are decomposed into main branch features $\boldsymbol{f}_l^{\text{main}}, \boldsymbol{f}_r^{\text{main}} \in \mathbb{R}^{\frac{N}{4} \times H_f \times W_f}$ and gating branch features $\boldsymbol{f}_l^{\text{gate}}, \boldsymbol{f}_r^{\text{gate}} \in \mathbb{R}^{\frac{N}{4} \times H_f \times W_f}$ along the channel dimension. Next, $\boldsymbol{f}_l^{\text{Global}}$ and $\boldsymbol{f}_r^{\text{Global}}$ are sequentially processed through a depthwise separable convolution layer and a SiLU activation function, and then undergo an information transformation in the Stereo 2D Selective Scan (discussed in Section 3.2.3) to obtain $\hat{\boldsymbol{f}}_l^{\text{main}}$ and $\hat{\boldsymbol{f}}_r^{\text{main}}$. Meanwhile, the gating features $\boldsymbol{f}_l^{\text{gate}}$ and $\boldsymbol{f}_r^{\text{gate}}$ are activated by the SiLU function to produce $\hat{\boldsymbol{f}}_l^{\text{gate}}$ and $\hat{\boldsymbol{f}}_r^{\text{gate}}$, which serve as spatial

importance maps indicating regions that require stronger information propagation. Finally, we use the stereo gating connection to further transfer the processed features as follows:

$$
\begin{aligned}
\hat{\boldsymbol{f}}_l^{\text{Global}} &= \text{Linear}(\text{Cat}(\hat{\boldsymbol{f}}_{l\to l}, \hat{\boldsymbol{f}}_{r\to l})) + \boldsymbol{f}_l^{\text{Global}}, \\
\hat{\boldsymbol{f}}_r^{\text{Global}} &= \text{Linear}(\text{Cat}(\hat{\boldsymbol{f}}_{r\to r}, \hat{\boldsymbol{f}}_{l\to r})) + \boldsymbol{f}_r^{\text{Global}},
\end{aligned}
\tag{4}
$$

where $\hat{\boldsymbol{f}}_{l\to l} = \hat{\boldsymbol{f}}_l^{\text{main}} \times \hat{\boldsymbol{f}}_l^{\text{gate}}, \hat{\boldsymbol{f}}_{r\to r} = \hat{\boldsymbol{f}}_r^{\text{main}} \times \hat{\boldsymbol{f}}_r^{\text{gate}}$ represent the important information in the current image that requires processing, $\hat{\boldsymbol{f}}_{r\to l} = \hat{\boldsymbol{f}}_r^{\text{main}} \times \hat{\boldsymbol{f}}_l^{\text{gate}}, \hat{\boldsymbol{f}}_{l\to r} = \hat{\boldsymbol{f}}_l^{\text{main}} \times \hat{\boldsymbol{f}}_r^{\text{gate}}$ represent the information from the other image that matches the perspective of the current image.

### 3.2.3 STEREO 2D SELECTIVE SCAN

In the selective state space, the input-dependent parameter matrix $C$ maps the hidden state $\boldsymbol{h}_t$ to the output, dynamically adjusting which features of the hidden state are amplified or suppressed based on the current input. Building on this concept, we introduce control information from the other view through matrix $C$ and propose a novel module called Stereo 2D Selective Scan. Specifically, following Vmamba (Liu et al., 2024b), we first unfold the image features $\hat{\boldsymbol{f}}_l^{\text{Global}}, \hat{\boldsymbol{f}}_r^{\text{Global}} \in \mathbb{R}^{\frac{N}{2} \times H_f \times W_f}$ into one-dimensional sequences $\boldsymbol{w}_l, \boldsymbol{w}_l \in \mathbb{R}^{\frac{N}{2} \times H_f W_f}$ through four-directional scanning. For a scanned feature in a specific direction, we first obtain the hidden states as follows:

$$
\begin{aligned}
A_l{'}, B_l{'} &= e^{\Delta_l A_l}, \Delta_l B_l, \quad h_l{}^t = A_l{'} h_l{}^{t-1} + B_l{'} w_l{}^t, \\
A_r{'}, B_r{'} &= e^{\Delta_r A_r}, \Delta_r B_r, \quad h_r{}^t = A_r{'} h_r{}^{t-1} + B_r{'} w_r{}^t,
\end{aligned}
\tag{5}
$$

Next, we perform a weighted summation of the control parameter $C$ from the other view using a learnable parameter $\alpha$, initially set to 0, and obtain the hidden states output:

$$
\begin{aligned}
v_l{}^t &= (C_l + \alpha C_r) h_l{}^t + D_l w_l{}^t, \\
v_r{}^t &= (C_r + \alpha C_l) h_r{}^t + D_r w_r{}^t,
\end{aligned}
\tag{6}
$$

where $w_l^t, w_r^t$ represent the input at time step $t$, and $v_l^t, v_r^t$ denote the selective scan output. In this way, we explicitly introduce information from the other view with negligible computational and storage overhead. Meanwhile, $\alpha$ is a learnable parameter, allowing the model to determine the amount of information to incorporate from the other perspective.

### 3.3 BI-DIRECTIONAL MULTI-REFERENCE ENTROPY MODEL

The spatial autoregressive entropy model significantly improves the performance of LIC but introduces prohibitive computational overhead. Recent single-image compression study (Jiang & Wang, 2023) proposes checkerboard-pattern multi-reference entropy models as a promising remedy. However, directly extending this approach to SIC is non-trivial, as it captures only intra-view priors and overlooks the critical inter-view dependencies inherent in SIC. This omission results in inaccurate probability estimation and compromises entropy coding performance. To address this challenge, we develop a novel bi-directional multi-reference entropy model based on our proposed Stereo VSSB, which facilitates effective inter-view contextual references and provides efficient fast coding speed.

As shown in Fig. 3, the proposed bi-directional multi-reference entropy model consists of intra-view prior prediction and inter-view prior prediction, which integrates the spatial-wise checkerboard context and channel-wise auto-regressive context. The conditional dependencies of our model are expressed as follows:

$$
\begin{aligned}
\text{\color{blue}{Intra-view}} \quad \text{\color{green}{Inter-view}} \\
p_{\hat{\boldsymbol{y}}_l^{ac}}(\hat{\boldsymbol{y}}_{l,i}^{ac} | \boxed{\color{blue}{\Phi_l^h, \Phi_{l,i}^{ch}, \Phi_{l,i}^{ter}}}, \boxed{\color{green}{\Phi_{l,i}^{iac}}}) &\sim \mathcal{N}(\boldsymbol{\mu}_l^{ac}, \boldsymbol{\sigma}_l^{2ac}), \\
p_{\hat{\boldsymbol{y}}_r^{ac}}(\hat{\boldsymbol{y}}_{r,i}^{ac} | \boxed{\color{blue}{\Phi_r^h, \Phi_{r,i}^{ch}, \Phi_{r,i}^{ter}}}, \boxed{\color{green}{\Phi_{r,i}^{iac}}}) &\sim \mathcal{N}(\boldsymbol{\mu}_r^{ac}, \boldsymbol{\sigma}_r^{2ac}), \\
p_{\hat{\boldsymbol{y}}_l^{na}}(\hat{\boldsymbol{y}}_{l,i}^{na} | \boxed{\color{blue}{\Phi_l^h, \Phi_{l,i}^{ch}, \Phi_{l,i}^{ter}, \Phi_{l,i}^{lc}, \Phi_{l,i}^{tra}}}, \boxed{\color{green}{\Phi_{l,i}^{inc}}}) &\sim \mathcal{N}(\boldsymbol{\mu}_l^{na}, \boldsymbol{\sigma}_l^{2na}), \\
p_{\hat{\boldsymbol{y}}_r^{na}}(\hat{\boldsymbol{y}}_{r,i}^{na} | \boxed{\color{blue}{\Phi_r^h, \Phi_{r,i}^{ch}, \Phi_{r,i}^{ter}, \Phi_{r,i}^{lc}, \Phi_{r,i}^{tra}}}, \boxed{\color{green}{\Phi_{r,i}^{inc}}}) &\sim \mathcal{N}(\boldsymbol{\mu}_r^{na}, \boldsymbol{\sigma}_r^{2na}), \\
\text{\color{blue}{Intra-view}} \quad \text{\color{green}{Inter-view}}
\end{aligned}
\right\} \text{Anchor} \\
\right\} \text{Non-anchor}
\tag{7}
$$

where $\hat{\boldsymbol{y}}_l^{ac}$ and $\hat{\boldsymbol{y}}_l^{na}$ denote the anchor and non-anchor elements of $\hat{\boldsymbol{y}}_l$, respectively, as shown in Fig. 3 (a). $i$ indicates the index of the channel slices. For the left-view priors, we use the anchor/nonanchor

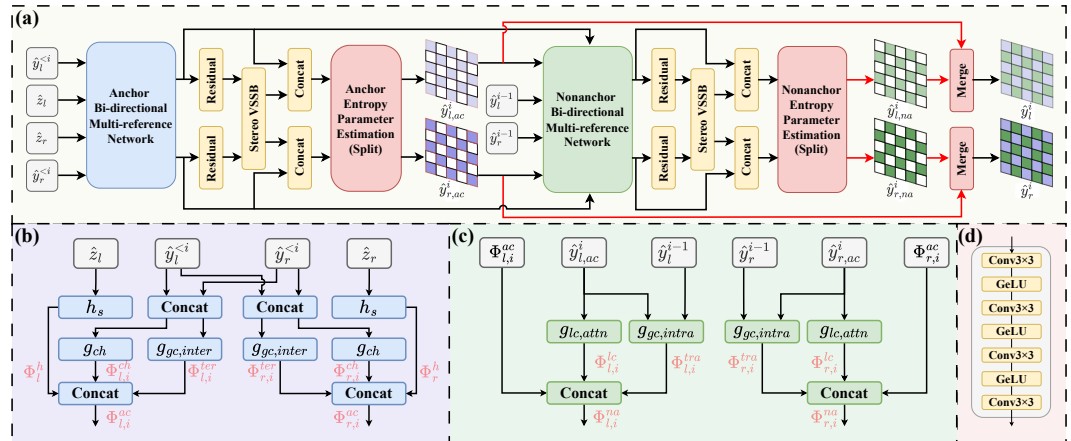

Figure 3: (a) The proposed bi-directional multi-reference entropy model. This figure illustrates the checkerboard-pattern entropy coding for a single slice. (b) Anchor bi-directional multi-reference network. (c) Nonanchor bi-directional multi-reference network. (d) Anchor/Nonanchor entropy parameter estimation network.

bi-directional multi-reference network in (Jiang & Wang, 2023) to generate a series of intra-view priors $\{\Phi_l^h, \Phi_{l,i}^{ch}, \Phi_{l,i}^{ter}, \Phi_{l,i}^{lc}, \Phi_{l,i}^{tra}\}$, which represent the hyperprior $\Phi_l^h$ from $\hat{z}_l$, the channel-wise auto-regressive prior $\Phi_{l,i}^{ch}$ from $\hat{y}_l^{<i}$, the local spatial context $\Phi_{l,i}^{lc}$ from $\hat{y}_{l,i}^{ac}$, the intra-slice global spatial context $\Phi_{l,i}^{tra}$ from $\{\hat{y}_l^{i-1}, \hat{y}_{l,i}^{ac}\}$, and the inter-slice global spatial context $\Phi_{l,i}^{ter}$ from $\hat{y}_l^{<i}$. $\Phi_{l,i}^{iac}$ and $\Phi_{l,i}^{inc}$ denote the proposed inter-view priors for $\hat{y}_l^{ac}$ and $\hat{y}_l^{na}$. This stereo multi-reference entropy model establish a strong prior for probability estimation, while the adopted checkerboard structure facilitates a faster processing than repeated spatial auto-regressive, which caters for both effectiveness and efficiency. We refer readers to Jiang & Wang (2023) for a detailed definition of intra-view priors.

Given the significant overlap and correlation between the left and right views, it is essential to introduce inter-view priors to establish the mutual interactions between views and progressively enhance the probability distribution estimation accuracy. Therefore, we apply our Stereo VSSB in Section 3.2.1 to generate the abundant inter-view priors $\{\Phi_{l,i}^{iac}, \Phi_{r,i}^{iac}\}$ and $\{\Phi_{l,i}^{ina}, \Phi_{r,i}^{ina}\}$ as follows:

$$\Phi_{l,i}^{iac}, \Phi_{r,i}^{iac} = V_i^{ac}(\Phi_{l,i}^{ac}, \Phi_{r,i}^{ac}),$$
$$\Phi_{l,i}^{ina}, \Phi_{r,i}^{ina} = V_i^{na}(\Phi_{l,i}^{na}, \Phi_{r,i}^{na}),$$
$$\Phi_{l,i}^{ac} = \Phi_l^h \oplus \Phi_{l,i}^{ch} \oplus \Phi_{l,i}^{ter}, \Phi_{l,i}^{na} = \Phi_{l,i}^{ac} \oplus \Phi_{l,i}^{lc} \oplus \Phi_{l,i}^{tra},$$
$$\Phi_{r,i}^{ac} = \Phi_r^h \oplus \Phi_{r,i}^{ch} \oplus \Phi_{r,i}^{ter}, \Phi_{r,i}^{na} = \Phi_{r,i}^{ac} \oplus \Phi_{r,i}^{lc} \oplus \Phi_{r,i}^{tra}, \tag{8}$$

where $V_i^{ac}$ and $V_i^{na}$ indicate the Stereo VSSB functions for the anchor and non-anchor views, respectively. $\oplus$ denotes the concatenation operation. Finally, we use intra-view priors $\{\Phi_{l,i}^{ac}, \Phi_{r,i}^{ac}, \Phi_{l,i}^{na}, \Phi_{r,i}^{na}\}$ and inter-view priors $\{\Phi_{l,i}^{iac}, \Phi_{r,i}^{iac}, \Phi_{l,i}^{ina}, \Phi_{r,i}^{ina}\}$ to effectively improve the estimation probabilities $\{p_{\hat{y}_l^{ac}}, p_{\hat{y}_r^{ac}}, p_{\hat{y}_l^{na}}, p_{\hat{y}_r^{na}}\}$.

### 3.4 Loss Function

Following the previous work, We employ the commonly used rate-distortion (RD) optimization framework to train our model. The overall loss function is defined as follows:

$$\mathcal{L} = \frac{1}{2}\sum_{l,r}(\lambda \cdot \mathcal{D}(\boldsymbol{x}_i, \hat{\boldsymbol{x}}_i) + (\mathcal{R}(\hat{\boldsymbol{y}}_i) + \mathcal{R}(\hat{\boldsymbol{z}}_i))), \tag{9}$$

where lagrange multiplier $\lambda$ controls the R-D tradeoff. $\mathcal{D}(\cdot, \cdot)$ denotes the distortion function as MSE or the MS-SSIM. $\mathcal{R}(\cdot)$ calculates bit-per-pixel using the entropy estimation as follows:

$$\mathcal{R}(\hat{\boldsymbol{y}}_l) = \Sigma_{i=0}^L(\mathcal{R}_{\hat{\boldsymbol{y}}_{l,i}^{ac}} + \mathcal{R}_{\hat{\boldsymbol{y}}_{l,i}^{na}}),$$
$$\mathcal{R}(\hat{\boldsymbol{y}}_r) = \Sigma_{i=0}^L(\mathcal{R}_{\hat{\boldsymbol{y}}_{r,i}^{ac}} + \mathcal{R}_{\hat{\boldsymbol{y}}_{r,i}^{na}}), \tag{10}$$

where $\mathcal{R}_{\hat{\boldsymbol{y}}_{l,i}^{ac}}, \mathcal{R}_{\hat{\boldsymbol{y}}_{r,i}^{ac}}$ and $\mathcal{R}_{\hat{\boldsymbol{y}}_{l,i}^{na}}, \mathcal{R}_{\hat{\boldsymbol{y}}_{r,i}^{na}}$ represent the anchor and non-anchor rates of the $i$-th slice for the left and right views, respectively. $L$ is the number of slices.

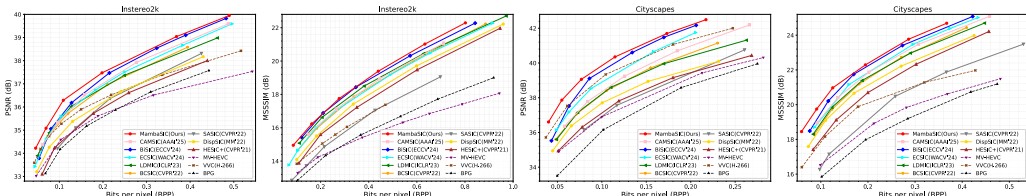

Figure 4: Rate-distortion curves in terms of PSNR and MS-SSIM on the InStereo2K and Cityscapes datasets. Our method outperforms pervious methods with a significant gap, benefiting from its superior stereo context fransfer method and entropy model.

## 4 EXPERIMENT

### 4.1 EXPERIMENTAL SETTINGS

**Datasets and Baselines.** We follow previous works (Liu et al., 2024c; Zhang et al., 2024b) to ensure a fair comparison and train our models on two widely used stereo image datasets, InStereo2K (Bao et al., 2020) and Cityscapes (Cordts et al., 2016). We compare MambaSIC with the hand-crafted coding standards BPG (Bellard, 2018), MV-HEVC (Tech et al., 2015) and H.266/VVC (Bross et al., 2021) as well as recent learning-based stereo image compression methods including HESIC+ (Deng et al., 2021), SASIC (Wödlinger et al., 2022), BCSIC (Lei et al., 2022), LDMIC (Zhang et al., 2023), ECSIC (Wödlinger et al., 2024), DispSIC (Zhai et al., 2022), BiSIC (Liu et al., 2024c) and CAMSIC (Zhang et al., 2024b). BPG encodes each view of the stereo pair independently, while MV-HEVC and H.266/VVC compress the left and right view images jointly.

**Implementation details.** We set the number of channels to $N = 128$ and $M = 320$, and configure the number of Stereo VSSB as $(n_1, n_2, n_3) = (1, 1, 1)$. We use Adam optimizer and optimize the network with the initial learning rate $1e - 4$ for 2M steps and then decreased to $1e - 5$ for another 0.8M steps and $1e - 6$ for the last 0.2M steps. For the first 1M steps, the batch size is set to 4, while for the remaining steps, it is set to 8. We set the rate-distortion trade-off multiplier in Eq. 9 as $\lambda \in \{0.0035, 0.0067, 0.0130, 0.0250, 0.0483, 0.0650\}$ for MSE loss and $\lambda \in \{4.58, 8.73, 16.64, 31.73, 60.5, 90.5\}$ for MS-SSIM loss.

### 4.2 EXPERIMENTAL RESULTS

**Compression Performance.** Fig. 4 and Table 1 reports the RD curves of all methods and BDBR results relative to BPG on InStereo2k and Cityscapes. MambaSIC reduces BDBR by 9.08% on InStereo2K and 8.94% on Cityscapes. Compared with unidirectional codecs, MambaSIC saves more

Table 1: BDBR$_{PSNR}$, BDBR$_{MSSSIM}$, BD-PSNR and BD-MSSSIM values of different compression methods. **Bold** indicates best results, and underlined values are the second-best ones.

| Method | InStereo2K | | | | Cityscapes | | | |
|---|---|---|---|---|---|---|---|---|
| | BD-PSNR | BDBR$_P$ | BD-MSSSIM | BDBR$_M$ | BD-PSNR | BDBR$_P$ | BD-MSSSIM | BDBR$_M$ |
| MVHEVC | 0.14dB | -7.69% | -0.13dB | 2.14% | 0.73dB | -18.02% | 0.62dB | -17.13% |
| VVC | 0.84dB | -35.31% | 0.92dB | -31.05% | 2.98dB | -56.25% | 1.92dB | -44.04% |
| HESIC+ | 0.39dB | -14.96% | 1.79dB | -43.22% | 0.99dB | -23.83% | 2.69dB | -50.79% |
| DispSIC | 0.68dB | -26.62% | 2.03dB | -47.89% | 1.47dB | -42.62% | 3.12dB | -59.06% |
| SASIC | 0.52dB | -18.40% | 0.74dB | -23.87% | 0.91dB | -21.47% | 1.38dB | -29.78% |
| BCSIC | 1.25dB | -41.22% | 2.45dB | -54.67% | 2.07dB | -42.62% | 3.50dB | -60.72% |
| LDMIC | 1.32dB | -41.95% | 2.71dB | -58.98% | 2.01dB | -41.92% | 3.55dB | -61.90% |
| ECSIC | 1.38dB | -43.71% | 2.44dB | -55.65% | 2.84dB | -52.06% | 3.93dB | -64.96% |
| BiSIC | 1.63dB | -48.07% | 2.95dB | -61.13% | 3.34dB | -57.49% | 4.21dB | -67.98% |
| CAMSIC | 1.46dB | -45.92% | 2.57dB | -55.20% | 2.28dB | -47.89% | 3.80dB | -65.16% |
| MambaSIC | **1.92dB** | **-57.15%** | **2.99dB** | **-62.89%** | **3.75dB** | **-66.43%** | **4.40dB** | **-72.45%** |

bits by observing a holistic view and mutually sharing features between stereo views, which facilitates removing redundancies in each view. Compared with bi-directional codecs, it achieves 15.93% to 9.08% extra BDBR reduction. This sugguests our entropy model aggregates more dependencies, and our stereo VSSB captures more inter-view correlations than 2D/3D convolutions.

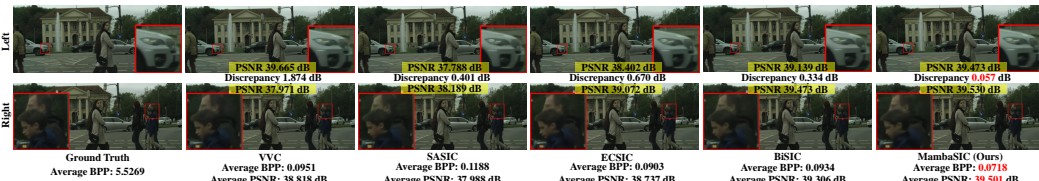

Figure 5: Qualitative comparison on reconstructed image across various codecs. Our MambaSIC achieves the lowest bit rate, the highest reconstruction quality, and the least PSNR discrepancy.

Fig. 5 presents a subjective comparison of our MambaSIC against various codecs on a stereo image pair from the Cityscapes dataset. It is demonstrated that our method not only exhibits superior reconstruction quality at a similar or lower bitrate, but also maintains consistent PSNR across stereo views due to its bi-directional architecture. In particular, the VVC codec compresses images in a predictive manner, resulting in an even larger PSNR gap of 1.874 dB between views. Meanwhile, the PSNR discrepancy between the two perspectives of BiSIC, which also has a bi-directional structure, is six times that of MambaSIC.

**Coding Latency.** We provide the coding latency analysis on the InStereo2K dataset in Table 2. All methods are tested on a single NVIDIA RTX 3090 GPU. Our method achieves the fastest encoding and decoding, being $62\times$ faster than BiSIC. This stems from our entropy model's adoption of a stereo-checkerboard instead of a spatial auto-regressive context, which simplifies the structure within the latents.

Table 2: Computational complexity of different methods on InStereo2K datasets.

| Method | Latency (s)↓ | | |
|---|---|---|---|
| | **Encode** | **Decode** | **Total** |
| SASIC | 4.7316 | 4.6964 | 9.4280 |
| BCSIC | 13.1341 | 29.9768 | 43.1109 |
| LDMIC | 11.3812 | 27.8496 | 39.2308 |
| ECSIC | 5.7061 | 5.3096 | 11.0157 |
| BiSIC | 32.8206 | 45.7868 | 78.6075 |
| CAMSIC | 0.9385 | 0.8116 | 1.7501 |
| MambaSIC | **0.6067** | **0.6558** | **1.2625** |

### 4.3 ABLATION STUDY

**Different cross-view matrix.** In Stereo 2DSS, we use matrix $C$ from both views for cross-view information transform. For comparison, we test using $B$, $\Delta$, and output $v$, as shown in Table 3. Adding cross-view control to $B$ gives decent results but is slightly worse than $C$, since $C$ is closer to the output in Eq. 5 and Eq. 6, giving it stronger influence on the final reconstruction. Using $\Delta$ or $v$ performs poorly, which further confirms that focusing on $C$ is the most effective design.

**Performance gain for efficacy and efficiency.** As shown in Table 4, we evaluate the speed performance of our proposed modules. V1 replaces 2D convolution and Stereo VSSB with 3D convolution and Mutual Attention Block from BiSIC. V2 replaces our entropy model with BiSIC's Cross-Dimensional Entropy Model. V3 only replaces Stereo VSSB with the Mutual Attention Block. Switching from a spatial autoregressive entropy model to a checkerboard model gives large speed gains.

Table 3: Comparison of cross-view matrix used in Stereo 2DSS. The first row is set as the anchor to measure BD-PSNR.

| Cross-view matrix | InStereo2K | Cityscapes |
|---|---|---|
| $C$ | 0 | 0 |
| $B$ | -0.058dB | -0.091dB |
| $\Delta$ | -0.100dB | -0.114dB |
| $v$ | -0.107dB | -0.012dB |

Table 4: Comparison of coding latency with different modules. We substitute the corresponding parts with the modules from BiSIC.

| Variant | Coding Latency (s) |
|---|---|
| Ours | 1.26 |
| (V1) w/ BiSIC codec backbone | 3.58 |
| (V2) w/ BiSIC entropy model | 75.19 |
| (V3) w/ BiSIC mutual attention | 2.64 |

Stereo VSSB runs faster than mutual attention, and 2D convolutions are more efficient than 3D convolutions. Overall, inter-view priors improve compression, the checkerboard entropy model boosts speed, and Stereo VSSB balances efficiency and rate-distortion well.

**Intra-module ablation.** We assess the effectiveness of each component in our stereo context transfer, with results shown in Table 5. Removing the cross view matrix (V1) increases BPP by 3.86% and 3.19% at the same PSNR. Removing the stereo gating connection (V2) raises BPP by 6.96% and 7.64%. Further replacing the entire stereo VSS block with a single-

Table 5: Ablation studies for different components. The first row is set as the anchor to measure BDBR on PSNR.

| Variant | InStereo2K | Cityscapes |
|---|---|---|
| Ours | 0% | 0% |
| (V1) w/o cross view matrix $\alpha C$ | 3.86% | 3.19% |
| (V2) w/o stereo gating connection | 6.98% | 7.64% |
| (V3) w single VSSB | 10.13% | 12.67% |
| (V4) w/o inter-view priors | 11.67% | 13.01% |
| (V5) w/ BCSIC entropy model | 9.79% | 10.11% |
| (V6) w/ BiSIC-fast entropy model | 8.39% | 6.16% |
| (V7) w/ BiSIC entropy model | 4.05% | 2.98% |
| (V8) w/ BCSIC Bi-CTM | 8.81% | 9.26% |
| (V9) w/ BiSIC Mutual Attention | 13.59% | 15.74% |

view version (V3) leads to a 10.13% and 12.67% BPP increase. Fig. 6 further demonstrates that the two branches in Stereo VSSB respectively enhance local texture information and global structural information. For more details, please refer to the section A.

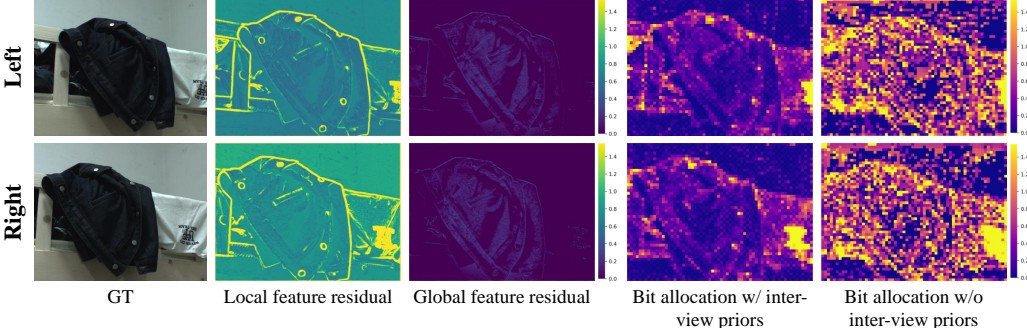

Figure 6: Examples from the InStereo2K demonstrate that the different paths in Stereo VSSB enhance local and global information, and incorporating inter-view priors leads to reduced bit allocation.

**Different entropy models.** We conduct ablation studies by replacing our entropy model with alternatives from BCSIC (V5), BiSIC (V6), and BiSIC-fast (V7), and by removing inter-view priors while using only the original model of MLIC++ (V4), as shown in Table 5. Compared with MLIC++, our model achieves a 13.01% bitrate reduction. As shown in Fig. 6, by incorporating inter-view priors, our method clearly allocates fewer bits. Compared with BCSIC and BiSIC variants, our entropy model better fuses left-right priors, further improving coding efficiency and reducing overhead.

**Inter-view fusion.** To evaluate the effectiveness of the proposed Stereo VSSB, we consider two baselines for comparisons. We replace Stereo VSSB with the mutual attention block in BiSIC (Liu et al., 2024c) and Bi-CTM in BCSIC (Lei et al., 2022) as variant V8 and V9. As shown in Table 5, Our proposed model outperforms all baselines by a large margin, which demonstrates its significance.

## 5 CONCLUSION

In this paper, we introduce MambaSIC, a novel stereo image compression framework differs fundamentally from previous CNN-based and attention-based approaches. To address inter-view redundancy, we introduce a Mamba-based stereo transfer module that leverages visual state-space modeling for efficient long-range dependency capture with linear complexity, enabling faster and richer latent representation. Furthermore, we develop a bidirectional multi-reference entropy model based on a checkerboard strategy and the proposed stereo transfer module, which achieves accurate probability estimation and faster entropy coding. Experiments demonstrate that MambaSIC outperforms state-of-the-art methods in both rate-distortion performance and speed, offering a practical solution for real-time and large-scale stereo compression tasks.

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

# A    MORE DETAILS ABOUT SECTION 4.3

Owing to space constraints, Section 4.3 presents a concise summary of the results. In this section, we conduct an in-depth analysis of each component.

**Intra-module ablation.** Variant V1 removes the cross-view control matrix $C$ and modulation parameter $\alpha$ in Stereo 2DSS. V2 builds on V1 by removing stereo gating, relying only on $\hat{f}_{l \to l}$ and $\hat{f}_{r \to r}$ without cross-view interaction—essentially reducing to the VSSL used in Vmamba Liu et al. (2024b). V3 extends V2 by eliminating the context transform, where each view's features are split by channel and independently processed through convolutional layers and VSSL, without any cross-view interaction. Among them, V3 shows the largest performance drop, while V1 shows the least, confirming the importance of each component.

We also ablate local and global modeling by removing the convolutional branch and retaining only the Stereo VSSL. This results in BDBR increases of 8.57% and 3.48% on two benchmarks, verifying the benefit of combining local and non-local features.

**Different Entropy Models.** In V4, we replace our entropy model with that of single image compression (Jiang & Wang, 2023), which only models multi intra-view priors for both anchor and non-anchor parts, without leveraging inter-view priors from the Stereo VSSB. Compared with our full model, V4 leads to bitrate increases of 11.67% and 13.01%, the most significant performance drop among all variants. These results underscore the importance of incorporating inter-view priors, which enable more accurate probability estimation and more efficient entropy coding. We also evaluate entropy models from state-of-the-art SIC methods (Lei et al., 2022; Liu et al., 2024c). As shown in Table 5, the proposed entropy model achieves better rate-distortion performance than baselines (V5,V6 and V7) This suggests that our model provides more accurate probability estimations, which in turn minimizes the coding overhead.

**Inter-view Fusion.** To evaluate the effectiveness of the proposed Stereo VSSB, we consider two baselines for comparisons. We replace the Stereo VSSB with the mutual attention block in BiSIC (Liu et al., 2024c) and Bi-CTM in BCSIC (Lei et al., 2022). We apologize for the mistake in Table 5—values for V8 and V9 were inadvertently swapped. The correct results should indicate that V8 yields bit rate increases of 13.59% and 15.74% on the two datasets, while V9 results in increases of 8.81% and 9.26%, respectively. We will correct this in the final version. As shown in Table 5, our proposed model outperforms all baselines by a large margin.

## B EXPERIMENTAL DETAILS

All training and testing settings strictly follow prior works (Liu et al., 2024c; Wödlinger et al., 2022; 2024; Zhang et al., 2023), to ensure fair comparisons. Specifically, each image in the InStereo2K dataset is pre-processed so that its dimensions are divisible by 64. For the Cityscapes dataset, rectification artifacts and the self-vehicle are removed by cropping 64 pixels from the top, 256 pixels from the bottom, and 128 pixels from each side of every image. During testing, we evaluate on images with resolutions of 1,024×832 from InStereo2K and 1,792×704 from Cityscapes.

For traditional codec baselines, BPG (Bellard, 2018) is evaluated using the YUV 4:4:4 format to retain high visual quality. HEVC and VVC are implemented using the JVET standard. Stereo image pairs are first converted into YUV 4:4:4 videos via ffmpeg, where the left image is encoded as an I-frame and the right as a P-frame. It is worth noting that MV-HEVC only supports YUV 4:2:0, which leads to degraded PSNR performance at higher bitrates. Additionally, we reproduce BCSIC (Lei et al., 2022) and evaluate it using the same image settings as in (Liu et al., 2024c; Wödlinger et al., 2022; 2024; Zhang et al., 2023), instead of the original 512×512 resolution used in (Lei et al., 2022), to ensure comparability. The original setup in (Lei et al., 2022) yields significantly lower RD values, hence we report all results under a unified and fair evaluation protocol.

## C ADDITIONAL VISUALIZATION RESULTS

We visualize the qualitative results in Fig.5, Fig.7, Fig.8, Fig.9, Fig.10 and Fig.11, to demonstrate the effectiveness of the proposed method compared with baseline models, including VVC(Bross et al., 2021), BCSIC (Lei et al., 2022), LDMIC (Zhang et al., 2023), SASIC (Wödlinger et al., 2022), ECSIC (Wödlinger et al., 2024), CAMSIC (Zhang et al., 2024b) and BiSIC (Liu et al., 2024c). Our proposed MambaISC achieves higher PSNR at lower BPP for both the left and right views, outperforming the compared methods. Besides, the reconstruction details and texture of BiSIC are closer to the ground truth. Notably, thanks to our bidirectional design, the image qualities of the left and right views remain consistent, effectively mitigating the imbalance issue often observed in unidirectional approaches. In contrast, VVC adopts a predictive compression framework where one view is encoded independently, and the other is generated based on the disparity between the predicted and actual views. This unidirectional approach results in a PSNR gap between stereo views. ECSIC compresses the right image using spatial context from the left image, yielding higher quality on the right view. SASIC uses the left image as a shift to assist the compression of the right image, which also results in a similar phenomenon. Compared with BiSIC, which also adopts a bidirectional structure, our method achieves a smaller PSNR discrepancy between views, indicating that the proposed Stereo VSSB is more effective than the mutual attention block in maintaining balanced reconstruction quality across views.

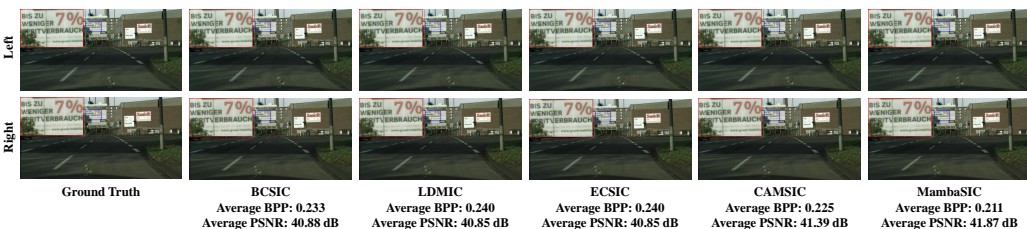

Figure 7: Qualitative comparison on reconstructed image across various codecs.

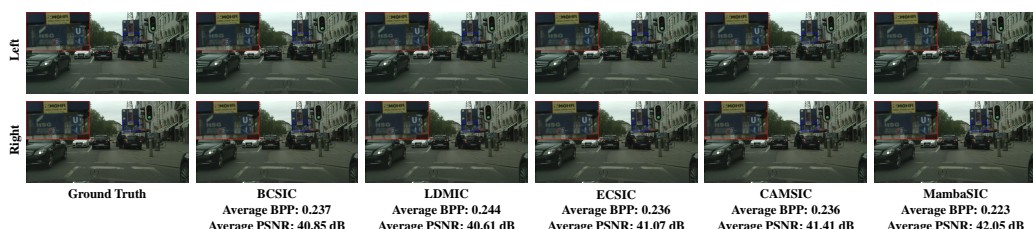

Figure 8: Qualitative comparison on reconstructed image across various codecs.

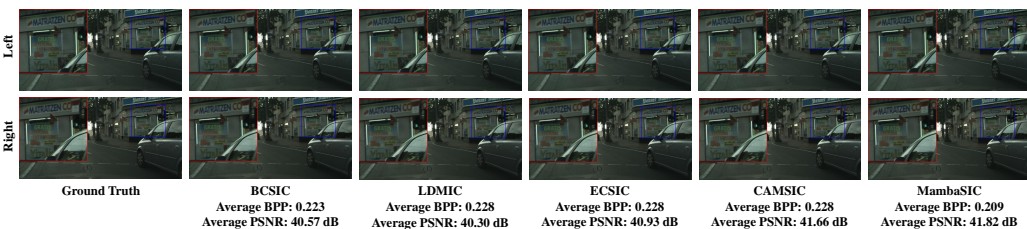

Figure 9: Qualitative comparison on reconstructed image across various codecs.

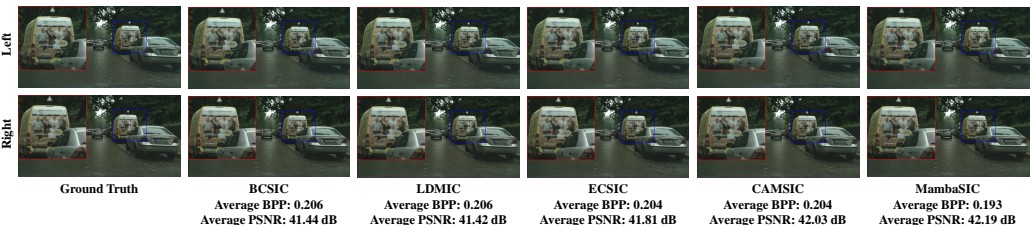

Figure 10: Qualitative comparison on reconstructed image across various codecs.

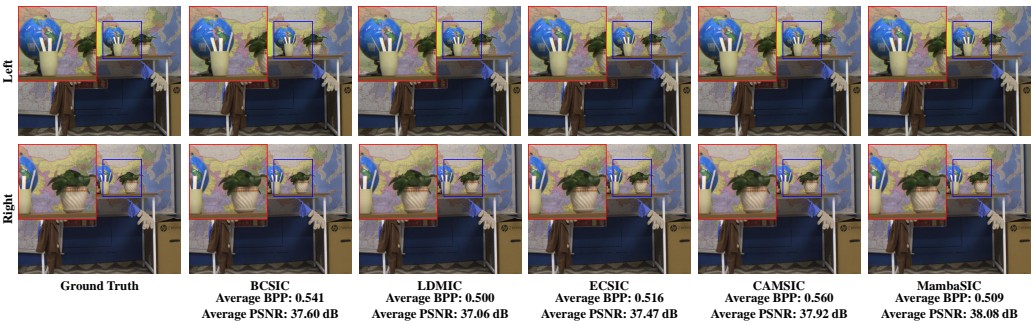

Figure 11: Qualitative comparison on reconstructed image across various codecs.

