# OpenReview forum: "MambaSIC: Mamba-based Stereo Image Compression with Bi-directional Multi-reference Entropy Model"
_ICLR.cc/2026/Conference — ICLR 2026 Conference Withdrawn Submission_

### Official Review · Reviewer_V835 · 2025-10-26

**Soundness:** 2
**Presentation:** 2
**Contribution:** 3
**Rating:** 6
**Confidence:** 3

**Summary:**

This paper proposes MambaSIC, a novel stereo image compression framework that leverages the Mamba-based Stereo Visual State Space Block (Stereo VSSB) to efficiently capture long-range inter-view correlations with linear complexity. It further introduces a bi-directional multi-reference entropy model that employs a checkerboard partition and inter-view priors for accurate probability estimation and fast entropy coding. Experiments on InStereo2K and Cityscapes show that MambaSIC achieves state-of-the-art rate–distortion performance.

**Strengths:**

1. The MambaSIC achieves both high compression efficiency and coding speed.
2. The paper is well-organized with clear figures.

**Weaknesses:**

1. The disparity between the left and right views can be significantly large. Within the proposed Stereo Visual State Space Block (Stereo VSSB), the straightforward concatenation of features from both views may introduce substantial irrelevant or noisy information. This unaligned fusion could contaminate the original feature representations and degrade their discriminative capacity, ultimately reducing the overall effectiveness of inter-view redundancy modeling.
2. The integration of the Mamba mechanism into stereo feature interaction is implemented in a rather direct and naive manner. The authors simply apply Mamba-based state-space modeling across two views without introducing any geometry-aware or adaptive fusion strategy. As a result, the conceptual novelty of the Stereo VSSB is limited, since it mainly reuses Mamba’s existing selective scan structure without substantial modification or theoretical adaptation to stereo correspondence.
3. Can the author visualize the feature map or attention-map in the Stereo VSSB to make it clear that Mamba does capture meaningful inter-view dependencies?

**Questions:**

See weaknesses.

---

### Official Review · Reviewer_cUHs · 2025-11-01

**Soundness:** 2
**Presentation:** 2
**Contribution:** 2
**Rating:** 2
**Confidence:** 4

**Summary:**

This paper proposes MambaSIC, a learning-based framework for stereo image compression (SIC) that leverages the Mamba architecture, specifically its State Space Model (SSM), to efficiently model long-range dependencies within and across stereo image views.  The core innovation is the Stereo Visual State Space Block, which is integrated into both the encoder and decoder. Experimental results show that MambaSIC achieves state-of-the-art performance, significantly outperforming existing hand-crafted (e.g., H.266/VVC) and learning-based (e.g., CAMSIC, BiSIC) SIC methods.

**Strengths:**

(1)	The paper includes a valuable ablation study that dissects the contribution of different components of the Stereo 2DSS module (e.g., cross-view control matrix, stereo gating, context transform).  This analysis helps to validate the design choices and shows the importance of each proposed element.

(2)	The reported performance gains are substantial and impressive.  The significant BDBR reduction compared to a wide range of strong baselines, including the recent CAMSIC and BiSIC.

**Weaknesses:**

(1)	From the visualization diagrams from Figure 7 to Figure 11, it is very difficult for me to notice the subjective visual differences among the various methods.

(2)	For the comparison of the practicality of the models, the author should provide more detailed FLOPs and Params of each model.

(3)	In my opinion, this paper mainly introduces the previously existing Mamba-based image compression [1]-[2] into stereo image compression and designs a dual-branch architecture. The VSS block is the original structure in VMamba [3], and this paper merely extends it to dual branches and adds structures such as Concat. Furthermore, the Checkboard entropy model has been very common in the field of image compression (whether unimodal or multimodal), sot the innovation of the proposed entropy model in this paper is insufficient. To sum up, I believe that the novelty and contribution of this paper are not sufficient to be published in ICLR.

[1] Zeng, Fanhu, et al. "MambaIC: State Space Models for High-Performance Learned Image Compression." Proceedings of the Computer Vision and Pattern Recognition Conference. 2025.

[2] Chen, Yunuo, et al. "CMIC: Content-Adaptive Mamba for Learned Image Compression." arXiv preprint arXiv:2508.02192 (2025).

[3] Liu, Yue, et al. "Vmamba: Visual state space model." Advances in neural information processing systems 37 (2024): 103031-103063.

**Questions:**

NA

---

### Official Review · Reviewer_PcJL · 2025-11-01

**Soundness:** 3
**Presentation:** 2
**Contribution:** 1
**Rating:** 4
**Confidence:** 5

**Summary:**

This paper proposes MambaSIC, a stereo image compression framework based on the Mamba state-space model. The authors design a Stereo Visual State Space Block to transfer contextual information across stereo views with linear computational complexity, and a bi-directional multi-reference entropy model that employs a checkerboard partitioning strategy for efficient and accurate entropy coding. Experimental results on InStereo2K and Cityscapes datasets show that MambaSIC achieves strong rate–distortion performance.

**Strengths:**

· The proposed method achieves excellent BD-BR results and significantly reduces coding latency, clearly outperforming existing stereo codecs.
· The proposed bidirectional design achieves the smallest PSNR discrepancy between stereo views.
· The integration of multiple components is well-executed, and the comprehensive evaluation demonstrates the method's effectiveness.

**Weaknesses:**

· The checkerboard partitioning strategy was first proposed in Checkerboard Context Model for Efficient Learned Image Compression and has since been widely adopted in learned image and 3D point cloud compression. The paper does not provide sufficient acknowledgment or discussion of these prior works, nor does it clearly explain what is new in its adaptation for stereo coding.
He D, Zheng Y, Sun B, et al. Checkerboard context model for efficient learned image compression[C]//Proceedings of the IEEE/CVF Conference on Computer Vision and Pattern Recognition. 2021: 14771-14780.
· The key innovations (Mamba integration and checkerboard-based entropy model) are extensions of known techniques, leading to relatively weak originality.
· The paper primarily demonstrates empirical improvements without offering deeper analysis of why the Mamba state-space formulation is more effective than attention-based alternatives in the stereo setting.

**Questions:**

· Could the authors more clearly delineate which aspects of their method represent genuine innovations versus adaptations of existing techniques? And did the authors attempt to integrate Mamba with other existing entropy structures ?
· Have the authors considered more innovative architectural variations beyond the relatively direct application of Mamba modules?
· Given the reliance on established techniques, what specific insights does this work provide that could guide future research in stereo compression?

---

### Official Review · Reviewer_EvsK · 2025-11-01

**Soundness:** 3
**Presentation:** 3
**Contribution:** 2
**Rating:** 4
**Confidence:** 4

**Summary:**

This paper presents MambaSIC, a stereo image compression framework that leverages Mamba-based visual state space blocks (stereo VSSB) to capture inter-view redundancy with linear complexity. The authors further propose a bi-directional multi-reference entropy model with checkerboard partitioning for efficient entropy coding. The method achieves state-of-the-art rate–distortion performance while being faster than existing stereo compression approaches.

**Strengths:**

1. The proposed method achieves strong empirical results. It reports a 57.15% BD-rate reduction over BPG on InStereo2K, outperforming BiSIC by 9.08%, while being faster than both BiSIC and CAMSIC. Moreover, it achieves the smallest inter-view PSNR discrepancy (0.057 dB vs. 0.334 dB for BiSIC), indicating that the model preserves cross-view consistency effectively and produces high-quality reconstructions.

2. By leveraging Mamba’s linear complexity and long-range modeling capability, the method provides an efficient way to capture inter-view dependencies.

3. The evaluation is comprehensive. The authors conduct thorough comparisons with over ten baselines, including both traditional codecs (BPG, MV-HEVC, VVC) and learning-based models, and evaluate across multiple benchmarks such as InStereo2K and Cityscapes. These extensive experiments demonstrate consistent improvements in both rate–distortion and runtime performance, lending strong credibility to the empirical claims.

**Weaknesses:**

1. While the integration of Mamba and checkerboard entropy modeling is effective, the paper’s conceptual novelty is limited. Most of the core components are borrowed or adapted from existing works, including Visual State Space models (VSSL, VMamba), four-directional scanning and checkerboard entropy modeling from MLIC++, and spatial partitioning strategies. As a result, the contribution appears primarily engineering-oriented.

2. The claim of achieving linear complexity over quadratic attention lacks rigorous theoretical or empirical support. Although Mamba inherently scales linearly with sequence length, the paper does not provide a formal FLOPs comparison or complexity analysis between the proposed stereo VSSB and mutual-attention mechanisms used in prior works such as BiSIC. Moreover, existing attention variants (e.g., linear or windowed attention) already achieve comparable efficiency, which weakens the argument that Mamba provides a unique complexity advantage in this context.

**Questions:**

Please refer to weaknesses.

---

### Note · Authors · 2025-11-12

I have read and agree with the venue's withdrawal policy on behalf of myself and my co-authors.